# The Inevitable Evasion? We Need More Than Detection to Combat Deepfake Audios.

## Abstract

This paper confronts the challenge of detecting increasingly sophisticated deepfake audio from advanced Text-to-Speech (TTS) systems with voice cloning. We posit that achieving high-accuracy, long-term detection of synthetic audio, particularly against motivated adversaries, is likely an unrealistic goal. This stance is supported by two primary observations. Firstly, the ongoing advancements in TTS and Synthetic Speech Detection (SSD) mirror an offline Generative Adversarial Network dynamic, with TTS as the generator and SSD as the discriminator. This suggests an eventual convergence towards synthetic speech that is nearly indistinguishable from human speech, making detection inherently challenging, if not impossible, especially as SSD development inherently lags behind TTS progress because SSD relies on TTS to generate training data. Secondly, current SSDs demonstrate a critical vulnerability to active, malicious evasion attacks, where the audio is carefully edited to bypass the target SSDs. Consequently, addressing deepfake audio demands a more systematic and multifaceted strategy, integrating approaches such as detection, legislative frameworks, watermarking technologies, robust enforcement mechanisms, and fostering cultural awareness.

## 1 Introduction

Text-to-speech (TTS) generation technologies are advancing rapidly. Modern systems [12, 3, 15, 21, 22, 26, 4, 5, 29, 7, 14, 18], readily available through open-source projects and commercial service APIs, now often include powerful zero-shot voice cloning capabilities (Table 1). This allows for the creation of convincing voice impersonations from mere seconds of audio, democratizing a technology once confined to specialized labs.

However, this accessibility brings profound ethical and societal risks. The potential for misuse is vast, ranging from sophisticated fraud and copyright infringement to the deliberate spread of misinformation via deepfaked voices of public figures / acquaintances. High-profile incidents, such as the use of an AI-generated voice clone of President Biden in illegal robocalls to disrupt the New Hampshire Presidential Primary Election [6], underscore the tangible threat and the urgent need for countermeasures.

Synthetic Speech Detectors (SSDs) [24, 23, 11], which distinguish between real and AI-generated speech using acoustic and linguistic cues, have emerged as the primary defense against such misuse. Yet, a critical question remains: Can these detectors reliably identify deepfake speech in practice? This paper, however, presents a more cautious, even pessimistic, outlook, positing that **the inherent efficacy of SSDs may be fundamentally limited by two intertwined and co-evolving dynamics: the co-evolving nature of SSDs and TTSs, and the perpetual adversarial contest between deepfake creators and SSDs**.

Firstly, the natural progression of TTS systems leads to generated audio becoming increasingly indistinguishable from genuine human speech, which consequently heightens the difficulty of detection for SSD. This has been repeatedly demonstrated [16]: more advanced TTS systems can produce outputs that readily circumvent SSDs trained on data from older, less sophisticated TTS systems. This inherent dynamic, where TTS technology continually evolves towards greater realism, suggests a trajectory towards a state where distinguishing generated audio from real human audio may become computationally prohibitive or even fundamentally infeasible, thereby necessitating watermarking technologies. Compounding this, SSD development inherently lags behind advancements in TTS technology because SSD systems fundamentally rely on access to data generated by new and improved TTS systems to learn their evolving characteristics and update their detection models. While an ideal approach might involve fostering a responsible innovation culture—whereby developers ensure a corresponding detector is feasible or developed before publicly releasing significantly more advanced TTS systems—such a synchronized framework presents substantial practical hurdles in today's fast-paced technological ecosystem, especially given the immense research interest and rapid innovation cycles in TTS development.

Second, current SSD systems often lack the necessary robustness for reliable, real-world deployment, a vulnerability highlighted by their performance degradation when faced with natural perturbations [28]. For example, variations in background noise, room acoustics, microphone characteristics, different audio compression techniques, or even the sheer diversity of human speech—encompassing different accents, ages, genders, and emotional states—can significantly alter the input features these detectors rely on, leading to an increase in false positives or false negatives. However, while existing research acknowledges that SSD performance degrades under such "natural" conditions like transcoding artifacts or background noise (a phenomenon known as test domain shift), this focus often overlooks a more critical vulnerability: an adversary will not rely on incidental distortions but will deliberately manipulate synthetic audio with the specific intent to deceive detection systems. The potential for success in such malicious scenarios is significantly higher, yet systematic research into these active attacks has been notably lacking. Consequently, this work takes the position that current state-of-the-art SSDs are demonstrably unreliable when confronted with dedicated adversarial attacks. We provide the first systematic study examining the vulnerability of leading open-source detectors to active malicious perturbations, investigating various attack scenarios from white-box attacks (full knowledge of the detector) to black-box and even transfer-based agnostic attacks (no direct access). Our evaluation considers not just the success rate of evasion but also the perceptual quality (stealthiness) of the attacked audio using both objective metrics and human assessment. Ultimately, these combined vulnerabilities to both environmental/data variations and the potential for targeted evasion, particularly demonstrated by our findings of signifi-

Table 1: Summary of SOTA TTS systems and their support for voice-cloning capability. ● means full support. ◑ means partial support. ○ means no support.

| System | Voice-cloning |
|---|---|
| **Commercial Platforms** | |
| ElevenLabs | ● |
| Descript | ● |
| Murf AI | ● |
| LOVO AI | ● |
| WellSaid | ◑ |
| Google Could TTS | ◑ |
| Amazon Polly | ○ |
| Azure AI Speech | ◑ |
| Resemble AI | ● |
| Play.ht | ● |
| Listnr.ai | ● |
| Fliki | ● |
| Synthesys | ● |
| NaturalReader | ○ |
| Speechify | ○ |
| Hume AI | ● |
| **Open-Source** | |
| Coqui TTS | ● |
| StyleTTS2 | ● |
| Bark | ● |
| Tortoise TTS | ● |
| Spark-TTS | ● |
| Kokoro TTS | ● |
| ChatTTS | ● |

cant security gaps, mean that current SSD systems cannot yet be considered consistently robust and argue for a fundamental rethinking of deepfake audio detection robustness in the face of determined adversaries.

## 2  Evolving TTS Naturally Evades Detection

### 2.1  The accuracy of SSD systems varies on different TTS systems

The accuracy of SSD systems is demonstrably not uniform when confronted with synthetic speech from different TTS systems. Instead, it exhibits significant variations contingent upon the specific TTS technology employed to generate the audio. An analysis [16] in Table 2 of average SSD performance across a suite of detection models reveals a considerable spread in accuracy based on the TTS source. For instance, when evaluating attacks from the PromptTTS2 [15] system, SSD models achieved an average accuracy of 80.00%. The average accuracy dropped to 71.87% for VALL-E [26], 60.51% for NaturalSpeech3 [22], and reached its lowest point at 47.27% for speech generated by OpenAI's TTS system. This wide spectrum of accuracy unequivocally underscores that the choice of TTS system is a critical factor influencing the accuracy of SSD systems.

Table 2: Averaged SSD performance on different TTS systems. Adapted from Table 4 in [16].

| System | Accuracy ($\uparrow$) | AUROC ($\uparrow$) | EER(%) ($\downarrow$) |
|---|---|---|---|
| PromptTTS2 | 0.8000 | 0.8372 | 20.000 |
| NaturalSpeech3 | 0.6051 | 0.6289 | 39.489 |
| VALL-E | 0.7187 | 0.7774 | 28.134 |
| VoiceBox | 0.7780 | 0.8294 | 22.203 |
| FlashSpeech | 0.6918 | 0.7468 | 30.817 |
| AudioGen | 0.6527 | 0.6654 | 34.455 |
| xTTS | 0.7376 | 0.8299 | 23.515 |
| Seed-TTS | 0.5711 | 0.5806 | 42.849 |
| OpenAI | 0.4727 | 0.3941 | 52.667 |

This observed variance in detection accuracy may also hint at a deeper, more intuitive challenge in the field of synthetic speech detection: the relationship between the perceived naturalness of synthetic speech and the difficulty of its detection. It is a prevailing hypothesis that as TTS technologies advance and produce speech that is increasingly indistinguishable from genuine human utterances—both in terms of acoustic quality and prosodic naturalness—they inherently become more formidable for SSD systems. **The underlying logic is that highly natural synthetic speech often minimizes or eliminates the subtle artifacts, unnatural cadences, or metallic reverberations that traditional SSD systems are trained to identify as tell-tale signs of spoofing.** While the provided dataset does not include explicit metrics for the naturalness of each listed TTS system, the accuracy patterns are thought-provoking when viewed through the lens of this hypothesis. The TTS systems against which the SSD suite performed relatively poorly, such as OpenAI, Seed-TTS [2], and NaturalSpeech3 [22], are often colloquially associated with producing highly natural and human-like speech. If these systems indeed generate more perceptually convincing audio, their lower corresponding SSD accuracy figures would lend support to the notion that increased naturalness directly correlates with a heightened challenge for detection. Conversely, systems against which SSDs achieved higher average accuracy, might be producing speech with more readily discernible synthetic markers, thus making them easier for detection algorithms to flag. This observation suggests a co-evolving dynamic between TTS and SSD technologies. As TTS systems evolve to create ever more realistic synthetic voices, SSD systems must concurrently advance to develop more sophisticated techniques capable of identifying these highly naturalistic spoofs.

### 2.2  TTS systems are evolving too fast for SSD to catch up

In the last section, we observed that SSD is likely to fail on more advanced TTS systems, and the co-evolving dynamic between TTS and SSD. The next question to answer naturally is:

**How fast is the development of TTS? Can we afford to always develop a reliable SSD for every TTS system before publicizing them? The anwer is unfortuantely negative.**

Table 3: Key TTS System Updates & New Releases in 2025

| System | Date | Key Details/Focus |
|---|---|---|
| Kokoro TTS | Jan 27, 2025 | Kokoro v1.0 model released. |
| Azure AI Speech | Feb 2025 | HD Neural TTS upgrade. |
| Spark-TTS | Feb 2025 | Paper on new LLM-based TTS model. |
| Hume AI | Feb 26, 2025 | Octave TTS launched. |
| Spark-TTS | Mar 19, 2025 | Spark-TTS-0.5B model details updated. |
| ChatTTS | Apr 29, 2025 | baseline models updated/released. |
| Google Cloud TTS | May 2025 | New controllable TTS models. |

The rapid evolution of TTS technology in 2025 is undeniable as shown in Table 3, with a continuous stream of new models and significant updates pushing the boundaries of synthetic speech. However, this progress presents a significant challenge for SSD. Detection systems inherently rely on being trained with data produced by the newest TTS systems. As new TTS models with novel architectures (like Spark-TTS's single-stream decoupled speech tokens [27]) or significantly improved quality emerge, existing detectors trained on older or different types of synthetic speech may become less effective. Research into defensive frameworks like SafeSpeech [30], which aims to protect audio before it can be used for high-quality synthesis by embedding imperceptible perturbations, highlights the difficulty of post-synthesis detection. The ability of modern TTS to achieve high-quality voice cloning from minimal samples, further amplified by LLMs generating human-like text prompts, means that detectors are in a constant state of catch-up. Each new breakthrough in TTS realism requires a corresponding update and retraining of detection models, creating an inherent lag.

If we take a longer-term perspective, the trajectory of TTS development is clearly aimed at achieving indistinguishability from real human voices. The emphasis across the industry is on creating voices that are not just intelligible but also convey genuine emotion, tone, and attitude. Systems like Hume AI's Octave TTS [1] explicitly aim to "understand what it's saying" to produce nuanced emotional expression, while models like StyleTTS 2 [17] strive "Towards Human-Level Text-to-Speech". The goal is to produce speech that is "smarter, faster, and more human than ever". This relentless pursuit of realism, while offering immense benefits in various applications, also intensifies the ethical concerns surrounding deepfakes and the potential for misuse, further underscoring how close synthetic speech is to becoming perceptually identical to human speech. Ultimately, achieving this indistinguishability will render detection impossible, necessitating proactive approaches like safeguards or watermarking—potentially at the cost of some quality—to prevent spoofing, alongside legislation to mandate these protective measures.

## 3   SSD Systems Are Vulnerable to Malicious Perturbations

It's already well known that current SSD systems are vulnerable to natural perturbations, such as different codecs [28]. In this section, we aim to answer the more radical question:

**Can deepfake audio be maliciously altered in ways nearly imperceptible to the human ear, but sufficient to bypass state-of-the-art detectors?**

Unlike previous research that focused on natural perturbations [20, 28], we consider a malicious attacker who deliberately optimizes the perturbation to evade detection. We examine this scenario under various levels of access to the detection systems, from having full knowledge (white-box), to partial knowledge (black-box), to no knowledge (agnostic).

### 3.1   Experiment Setup

We train four SOTA open-source SSDs from scratch: AASIST [11], AASIST-L [11], RawNet2 [24] and RawGATST [23] on ASVSpoof2019-LA train split [25]. Their equal error rates (EERs) without attacks on ASVSpoof2019-LA test split are reported in Table 4, and closely match the reported numbers in their original papers.

Table 4: Baseline EERs of SSDs on the ASVSpoof2019-LA test split without attacks.

| AASIST | AASIST-L | RawNet2 | RawGATST |
|--------|----------|---------|----------|
| 0.83%  | 0.99%    | 4.88%   | 3.29%    |

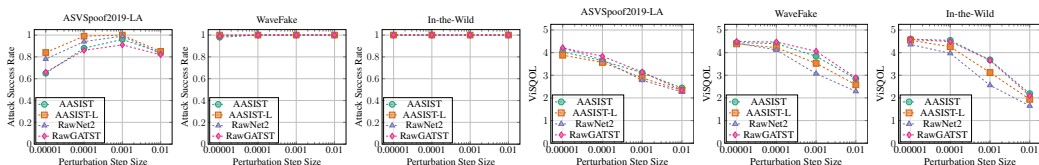

Figure 1: Attack success rate and ViSQOL vs. perturbation step size in PGD.

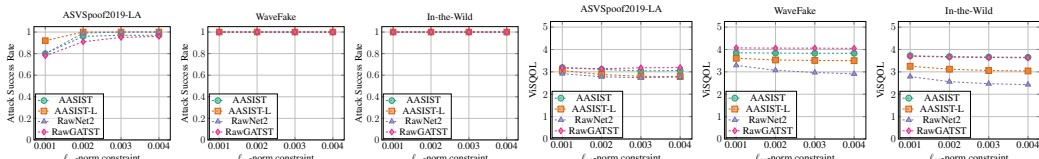

Figure 2: Attack success rate and ViSQOL vs. $\ell_\infty$-norm constraint in PGD.

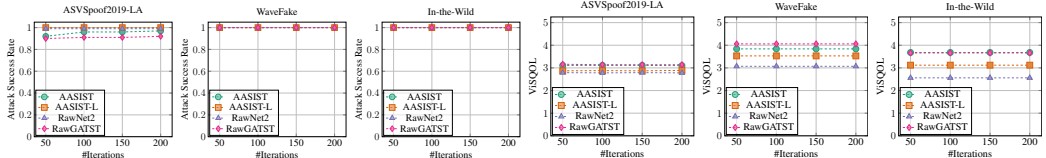

Figure 3: Attack success rate and ViSQOL vs. PGD #iterations across red-team sets.

We launch attacks on three synthetic datasets: ASVSpoof2019-LA test split [25], WaveFake [8] and In-the-wild [20]. In consideration of compute resources, we randomly sub-sample 100 examples from each dataset for the attacks.

We use the attack success rate (*i.e.* the ratio of attacked examples bypassing the target detector) to measure the effectiveness of the attacks. To ensure the attack does not degrade audio quality, we use both VisQOL [10] and human ratings to confirm that the attacked audio still sounds similar to the original synthetic audio, which we refer to as "stealthiness".

### 3.2 White-box Attack

We first study white-box attack, where the adversary has full access to the model. We choose two white-box attacks: Projected Gradient Descent (PGD) [19] and I-FGSM [13].

**Projected Gradient Descent:** PGD crafts adversarial examples by iteratively taking small steps in the direction that maximizes the model's error, while projecting the perturbed example back within a certain boundary around the original input to maintain a balance between attack success rate and stealthiness.

PGD has three major hyper-parameters: perturbation step size, $\ell_\infty$-norm constraint, and the number of iterations. We conduct hyper-parameter search and summarize the results in Figure 1, 2, and 3.

In Figure 1, we can tell that on WaveFake and In-the-wild, the attack success rate is almost always 100% while on ASVSpoof2019-LA test the attack success rate hovers between 60% and 100% depending on the learning rate used. This reflects the fact that the detectors are more robust on test data generated by the same TTS systems as the training data (*i.e.* in-domain data), but are still vulnerable under white-box attacks with a few steps of hyper-parameter search. On the other hand, VisQOL scores keep decreasing as the perturbation step size grows. Usually, VisQOL score above 3.0 is considered reasonable quality. Thus, there exists a sweet spot of perturbation step size striking balance between attack effectiveness and stealthiness.

Table 5: Human ratings of speaker similarity between the original and PGD attacked audio.

|  | ASVspoof | WaveFake | In-the-wild |
|---|---|---|---|
| AASIST | $0.970 \pm 0.063$ | $0.971 \pm 0.046$ | $0.985 \pm 0.030$ |
| AASIST-L | $0.979 \pm 0.037$ | $0.979 \pm 0.045$ | $0.975 \pm 0.036$ |
| RawNet2 | $0.971 \pm 0.077$ | $1.000 \pm 0.000$ | $0.967 \pm 0.063$ |
| RawGATST | $0.997 \pm 0.008$ | $0.986 \pm 0.030$ | $0.997 \pm 0.008$ |

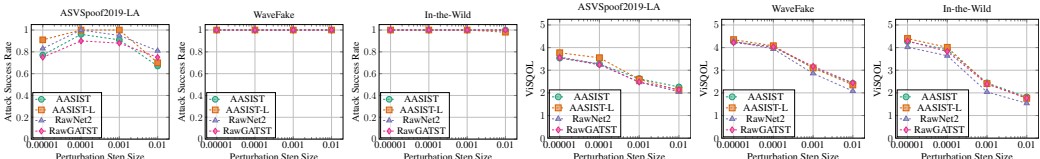

Figure 4: Attack success rate and ViSQOL vs. perturbation step size in I-FGSM.

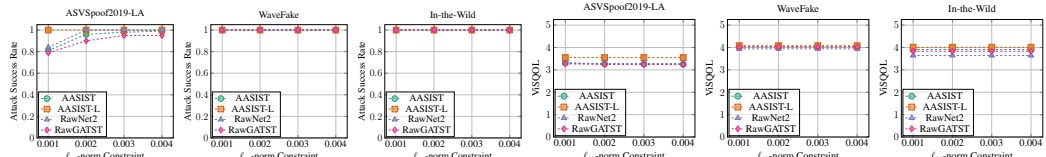

Figure 5: Attack success rate and ViSQOL vs. $\ell_\infty$-norm constraint in I-FGSM.

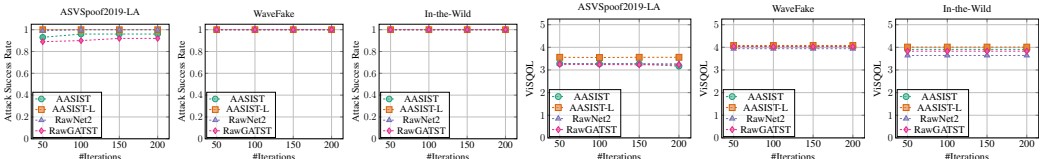

Figure 6: Attack success rate and ViSQOL vs. I-FGSM #iterations.

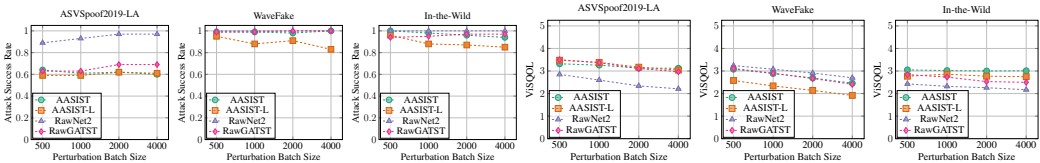

Figure 7: Attack success rate and ViSQOL vs. SimBA perturbation batch size.

In Figure 2, the observation of SSDs being more robust on ASVSpoof2019-LA test holds true. However, we observe that the VisQOL scores are pretty consistent despite the changing $\ell_\infty$-norm constraint, which says that audio quality is insensitive to $\ell_\infty$-norm constraint within a certain range.

Figure 3 shows that white-box attacks are efficient, reaching maximum attack success rates and stable VisQOL scores after just 50 iterations.

We also collect human ratings on whether the PGD-attacked audio with the best hyper-parameter combination sounds like the original synthetic audio, and the results are summarized in Table 5. We can see that most human raters think the two audio sound like the same person, underscoring the potential threat of using the attacked audio for impersonation.

**Iterative Fast Gradient Sign Method:** I-FGSM only differs from PGD in that it only uses the sign of the gradient to perturb the input audio. It shares the same set of hyper-parameters as PGD, for which the grid search results are summarized in Figure 4, 5, and 6 and human ratings are summarized in Table 6, and the findings are similar to PGD.

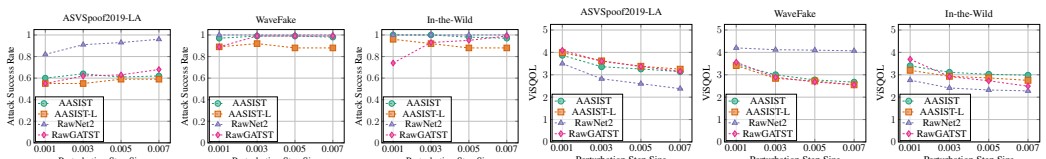

Figure 8: Attack success rate and ViSQOL vs. SimBA perturbation step size across datasets.

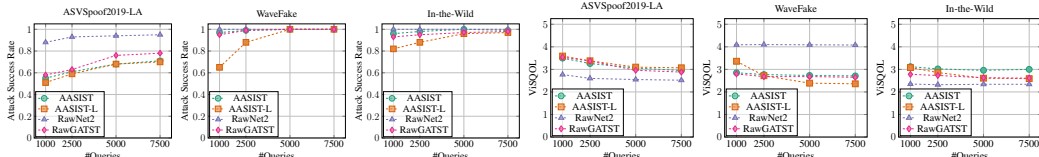

Figure 9: Attack success rate and ViSQOL vs. SimBA #queries across datasets.

## 3.3 Black-box Attack

For black-box attack, we choose the Simple Black Box Attack (SimBA) [9]. SimBA perturbs the input audio randomly and observes whether the prediction confidence score for "fake" class decreases or increases. If the confidence score decreases, SimBA will keep the perturbation. Otherwise, SimBA will try adding perturbation in the opposite direction and decide whether to keep or discard the perturbation just as above. SimBA iteratively perturbs the input until the SSD is successfully bypassed or the budget of queries/iterations is used up.

SimBA has three hyper-parameters: perturbation batch size, perturbation step size, and the number of queries. Perturbation batch size decides how many timesteps are perturbed in each query, while perturbation step size decides which long one perturbation step on one timestep can be. The hyper-parameter search results are summarized in Figure 7, 8, and 9.

In Figure 7, we observe that on ASVSpoof2019 test, RawNet2, the least capable SSD model is still broken almost 100% but all the other 3 models are only broken 60% of all the tested examples. This draws a positive correlation between model capability and robustness. On WaveFake and In-the-Wild, all SSDs are broken more than 90% of the time, which confirms the previous observation that current SSD models are brittle when facing synthetic audio from TTS systems never seen during training.

Also in Figure 7, 8, and 9, we observe that ASSIST-L is the most robust model consistently, which is surprising because it's the smallest model within the 4 (See [11] for the size of these models.). This observation aligns with the principle of Occam's razor, which suggests that simpler models often generalize better. A potential explanation could lie in the raggedness of the decision boundaries. Larger models, with their increased complexity, might create more intricate and potentially overfit decision boundaries. In contrast, ASSIST-L, being smaller, may form smoother decision boundaries, leading to better generalization and robustness against perturbations.

Human ratings of audio similarity is summarized in Table 7. Again the attacked audio sound highly similar to the original ones to human ears.

## 3.4 Agnostic Attack: Transferability of Above Attacks

The above attacks all assume different levels of access to the SSD model which might not be accessible in practice. As a result, we want to understand whether the above attacks are transferrable: Can a successfully attacked example on one model transfer to a different model? If this is true, then the adversary can craft a proxy model themselves, attack it, and expect it to bypass the real SSD as well.

Table 6: Human ratings of speaker similarity between the original and I-FGSM attacked audio.

|  | ASVspoof | WaveFake | In-the-wild |
|---|---|---|---|
| AASIST | $0.984 \pm 0.020$ | $0.960 \pm 0.052$ | $0.985 \pm 0.024$ |
| AASIST-L | $0.987 \pm 0.022$ | $0.986 \pm 0.023$ | $0.967 \pm 0.054$ |
| RawNet2 | $0.980 \pm 0.040$ | $1.000 \pm 0.000$ | $0.991 \pm 0.012$ |
| RawGATST | $0.989 \pm 0.024$ | $0.858 \pm 0.141$ | $0.985 \pm 0.024$ |

Table 7: Human ratings of speaker similarity between the original and simBA attacked audio.

| | ASVspoof | WaveFake | In-the-wild |
|---|---|---|---|
| AASIST | $0.984 \pm 0.020$ | $0.960 \pm 0.052$ | $0.985 \pm 0.024$ |
| AASIST-L | $0.987 \pm 0.022$ | $0.986 \pm 0.023$ | $0.967 \pm 0.054$ |
| RawNet2 | $0.980 \pm 0.040$ | $1.000 \pm 0.000$ | $0.991 \pm 0.012$ |
| RawGATST | $0.989 \pm 0.024$ | $0.858 \pm 0.141$ | $0.985 \pm 0.024$ |

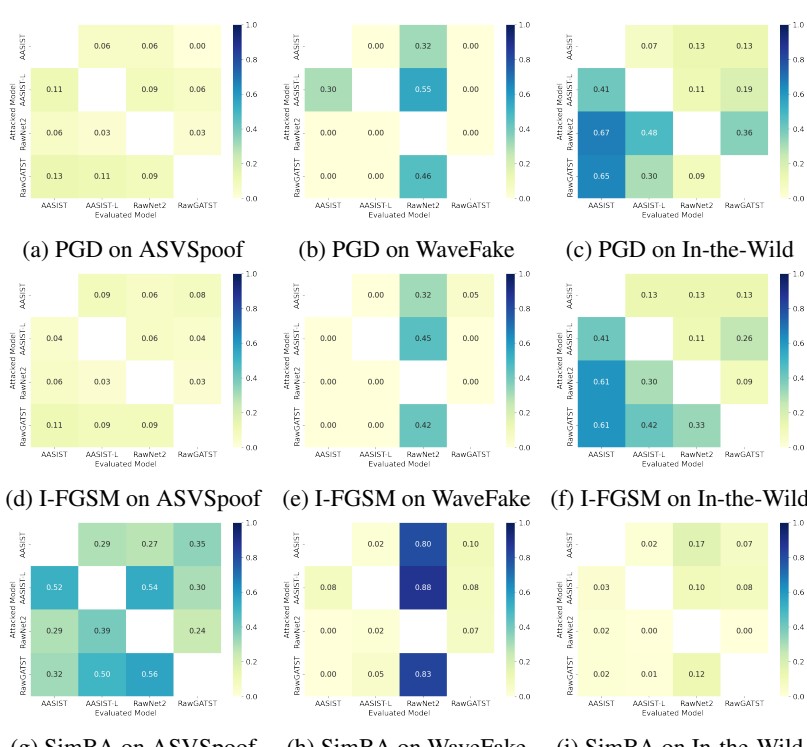

(a) PGD on ASVSpoof    (b) PGD on WaveFake    (c) PGD on In-the-Wild

(d) I-FGSM on ASVSpoof    (e) I-FGSM on WaveFake    (f) I-FGSM on In-the-Wild

(g) SimBA on ASVSpoof    (h) SimBA on WaveFake    (i) SimBA on In-the-Wild

Figure 10: Transferability of attacks on different datasets.

The results are summarized in Figure 10. First, we find that on out-of-domain data, some SSDs are extremely vulnerable. For example, on WaveFake, RawNet2 is extremely vulnerable under all attacks; on In-the-wild, ASSIST and AASIST-L are more vulnerable than the other two models. Second, we find that on in-domain data, black-box attacks are much more transferrable than white-box attacks. This is because 1) black-box attacks tend to add larger perturbation than white-box attacks; 2) the SSDs' decision borders are alike for in-domain data. Thirdly, we also observe high similarity between the transferrability heatmap between PGD and I-FGSM, which might be due to different white-box attacks taking gradient paths in similar directions despite small differences.

# 4 Conclusions and Open Problems

This paper has argued that the pursuit of high-accuracy, long-term detection of sophisticated deepfake audio, particularly in the face of motivated and adaptive adversaries, is an increasingly challenging, and perhaps ultimately unrealistic, endeavor. This position is supported by two primary lines of reasoning.

Firstly, the rapid and continuous advancements in TTS technologies are fundamentally driving synthetic audio towards a point of near-indistinguishability from genuine human speech. This dynamic mirrors an adversarial relationship where TTS generators constantly improve, making the task for SSD discriminators progressively harder. The inherent lag in SSD development, which relies on access to newer TTS outputs for training, further exacerbates this challenge.

Secondly, our findings, along with emerging research, demonstrate a critical vulnerability of current state-of-the-art SSDs to deliberate adversarial perturbations. We have shown that it is possible to introduce subtle, often imperceptible, modifications to synthetic audio that can successfully deceive leading detection models across various attack scenarios (white-box, black-box, and transfer-based). This susceptibility, even when audio quality is maintained, underscores the fragility of current detection paradigms against determined attackers.

Therefore, while detection remains a component of the response, relying on it as the primary or sole defense against the misuse of deepfake audio is a precarious strategy. The "arms race" is heavily skewed, and the evidence suggests that perfect, or even consistently reliable, detection is unlikely to be a stable long-term solution.

The challenges highlighted in this paper point to several critical open problems and avenues for future research:

**Proactive Defense Mechanisms:** What are the most effective and scalable methods for audio watermarking (both perceptible and imperceptible) that are resilient to removal or alteration? Can "audio immune systems" or pre-emptive perturbation techniques (akin to SafeSpeech [30]) be developed to make original audio recordings inherently more difficult to clone or manipulate convincingly?

**Understanding and Countering Adversarial Strategies:** Further systematic study is needed on the evolving tactics of adversaries in creating and deploying deepfakes, including more sophisticated attack algorithms and their transferability. How can we develop dynamic defense mechanisms that can adapt to new attack vectors in real-time?

**Human Perception and Media Forensics:** How do humans perceive and differentiate highly realistic deepfake audio from genuine speech, and can these perceptual cues be leveraged for detection? How can we improve media literacy to equip individuals with the skills to critically evaluate audio content? Developing reliable and accessible forensic tools for verifying audio authenticity remains a key challenge.

**Ethical Frameworks and Responsible Innovation:** How can we foster a culture of responsible innovation within the TTS development community, encouraging the proactive consideration of misuse and the integration of safeguards? What ethical guidelines and standards should govern the development, deployment, and accessibility of powerful voice synthesis technologies?

**Attribution and Legal Recourse:** Developing techniques for attributing the source or creator of a deepfake audio clip is crucial for accountability. How can legal and regulatory frameworks be adapted to effectively address the creation and malicious dissemination of deepfake audio while protecting freedom of expression?

## 5   Alternative View

While this paper posits a pessimistic outlook on the long-term efficacy of SSD systems, an alternative perspective holds that the "arms race" between TTS generation and SSD may not inevitably lead to the obsolescence of detection. This view is predicated on several possibilities:

1. Current SSDs primarily analyze acoustic and linguistic cues to distinguish between real and AI-generated speech. However, future research could unveil entirely new categories of tell-tale signs inherent to synthetically generated audio. These might be exceptionally difficult for TTS models to mimic or for adversaries to successfully perturb.

2. The trajectory of TTS development aims for synthetic voices that are virtually indistinguishable from human speech. However, human speech is the product of incredibly complex biological, physiological, and cognitive processes. Achieving true indistinguishability – capturing every nuance, subtle imperfection, and the inherent variability of genuine human vocal production – might be a far more intractable problem for AI than currently anticipated.

3. The paper notes that SSD development inherently lags behind TTS advancements because SSD systems rely on data from new TTS systems for training. However, a more concerted and perhaps preemptive research effort in SSD could shift the dynamic.

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
