# OpenReview forum: "Detection alone is insufficient to mitigate the harm by deepfake audio"
_NeurIPS.cc/2025/Position_Paper_Track — Submitted to NeurIPS 2025 Position Paper Track_

### Official Review · Reviewer_TLox · 2025-08-07

**Significance:** 2
**Presentation:** 3
**Rating:** 6
**Confidence:** 2

**Summary:**

This paper argues that relying solely on synthetic speech detection (SSD) to combat deepfake audio is a flawed and ultimately unsustainable strategy. The authors contend that TTS technology, which generates synthetic audio, is advancing at a rapid pace and is consistently outpacing SSD technology. This is because SSDs rely on data from new TTS systems to train their models, creating an inherent lag. As TTS systems produce increasingly human-like speech, the subtle artifacts that SSDs are trained to detect are minimized or eliminated, making detection more difficult.

The vulnerability of current SSDs to malicious adversarial attacks. The paper presents a systematic study demonstrating that SSDs can be evaded by deliberately perturbing the synthetic audio. These perturbations successfully deceive leading detection models, highlighting the fragility of current detection paradigms against determined attackers.

**Strengths:**

* Strong, Evidence-Based Argument: detection alone that is insufficient is well-supported by both a logical, theoretical framework (the TTS-SSD "arms race") and a concrete, experimental one (the adversarial attack study).
* Comprehensive Experimental Design: The study on adversarial attacks is a significant contribution. It systematically evaluates SSD vulnerabilities across different attack scenarios and on various datasets, providing concrete numbers and supporting data. The Attack Success Rate, VisQOL, and human ratings add credibility to the claim that the attacks are "stealthy."
* Actionable Recommendations: The conclusion moves beyond simply identifying the problem and offers a range of actionable recommendations for future research and policy, from proactive defense mechanisms like watermarking to ethical frameworks and legal recourse.

**Weaknesses:**

* Lack of Detail on Human Ratings: While Tables 5 and 6 summarize human ratings, the paper provides minimal information about the methodology. Details such as the number of raters, their background, etc., would be crucial for a thorough evaluation.
* Limited Discussion on Watermarking: The paper frequently suggests watermarking as a key part of the solution but provides very little detail on what this would entail.

**Questions:**

Why is the title different from the one submitted by the openreview system?

**Alternative Position:**

Yes, and alternative positions are well-considered and named but not addressed

**Author Identification:**

No.

**Context:**

3

**Discussion:**

3

**Ethics:**

["NO or VERY MINOR ethics concerns only"]

**Position:**

Yes, the paper argues for or against a position related to machine learning.

**Support:**

3

**Thoroughness:**

4

---

### Official Review · Reviewer_scPU · 2025-08-08

**Significance:** 3
**Presentation:** 3
**Rating:** 7
**Confidence:** 3

**Summary:**

This paper argues that reliable detection of deepfake audio is likely unrealistic in the long term. It supports this through two main findings: 1) TTS systems are evolving faster than detection systems can keep up, and 2) current detection systems are vulnerable to malicious attacks that can bypass them while maintaining audio quality. The paper demonstrates these vulnerabilities through experiments with different attack methods and concludes that detection alone cannot be the solution to deepfake audio threats.

**Strengths:**

- Position is clear -- modern detection alone is insufficient for combating deepfake audio
- Critical security challenge as AI gets better and more pervasive
- Good coverage of literature in text to speech and synthetic speech detection
- Impressive set of experiments that demonstrate that modern detection systems are vulnerable to various attacks, especially for a motivated adversarial attacker
- Evaluation procedure considers human ratings

**Weaknesses:**

- The position is quite narrow given that the same problem translates to text and video as well. The paper can take a more expansive stance.
- It is unclear if the detection still fails when multiple methods are used in an ensemble.
- What if there is a confidence score attached to the prediction, and the reader can assess if they want to further examine the source based on the risk level? Binary yes/no is too blunt

**Questions:**

- Have you analyzed whether these attacks leave other detectable artifacts in the frequency domain or in other audio characteristics not currently used by detection systems?
- Given that your results suggest detection alone is insufficient, could you elaborate on how the proposed alternative solutions (watermarking, legislation, etc.) might interact with or complement detection approaches?
- The paper takes a strong stance that detection alone is insufficient, yet many other security domains (like spam detection or malware detection) continue to rely primarily on detection-based approaches. What makes deepfake audio fundamentally different from these other security challenges?

**Alternative Position:**

Yes, and alternative positions are well-considered and named but not addressed

**Author Identification:**

No.

**Context:**

4

**Discussion:**

4

**Ethics:**

["NO or VERY MINOR ethics concerns only"]

**Position:**

Yes, the paper argues for or against a position related to machine learning.

**Support:**

4

**Thoroughness:**

3

---

### Official Review · Reviewer_iSXe · 2025-08-30

**Significance:** 3
**Presentation:** 3
**Rating:** 7
**Confidence:** 3

**Summary:**

This paper advocates that achieving high-accuracy, long-term detection of synthetic audio is likely an unrealistic goal. This position is backed up by two observations. The first observation is that TTS and SSD are usually developing together, particularly SSD lags behind TTS due to relying on the newest TTS as training data. The second observation is that SSD is vulnerable to active attacks, meaning that the synthetic audio is deliberately edited to bypass the target SSDs; and the authors assessed the attack success rate to measure the effectiveness of the attacks under three scenarios: white-box, black-box, and agnostic.

**Strengths:**

This paper addresses a critical and rapidly evolving topic – the robustness of SSD amid the fast development in TTS technologies, which has significant societal implications. Specifically, it identifies two main issues: the rapid evolution of TTS outpacing SSD development, and the vulnerability of SSDs to deliberate adversarial perturbations. Additionally, it suggests possible solutions to these challenges.

The study systematically tests four SOTA SSD models across diverse attack scenarios (white-box, black-box and transfer-based agnostic attacks), combining objective metrics and human ratings. This systematic assessment strengthens the validity of findings.

Moreover, this paper also presents alternative views that emphasize ongoing research potential, maintaining a balanced view rather than one-sided pessimism.

**Weaknesses:**

While human ratings are included, the paper could elaborate on the methodology to strengthen claims about stealthiness, such as number of raters, their demographic backgrounds and the specific criteria used to assess similarity.

The paper briefly touches on future directions, such as legislative frameworks and watermarking technologies, but provides limited concrete suggestions. Expanding on these could enhance the discussion.

**Questions:**

1. Could you please elaborate on the human rating methodology? Specifically, how many raters were involved, and what were their backgrounds? This would help better assess the reliability of the stealthiness claims.

2. Given the demonstrated vulnerability of SSDs, do you have any idea for short-term promising solutions, before watermarking or legislative measures become widely adopted?

**Alternative Position:**

Yes, and alternative positions are well-considered and named but not addressed

**Author Identification:**

No.

**Context:**

4

**Discussion:**

3

**Ethics:**

["NO or VERY MINOR ethics concerns only"]

**Position:**

Yes, the paper argues for or against a position related to machine learning.

**Support:**

4

**Thoroughness:**

3

---

### Note · Authors · 2025-09-01

**1-11 Submit Again:**

Definitely yes

**1-1 Submission Process:**

5

**1-2 Next Year:**

It would be interesting to have a few dedicated themes so we would be able to see different / complementary positions on the same topic. This can also foster discussion and collaboration in the chosen areas.

**1-3 Future Development:**

Publish a summary report of the session's key debates or invite top papers to a special journal issue to extend the impact beyond the conference.

**1-4 Interest:**

["Panel discussions with other position paper authors", "Structured debates on controversial topics", "Workshops for developing position papers", "Mentorship programs for early-career researchers"]

**1-5 Thoughtful:**

10

**1-6 Supportive:**

10

**1-7 Technical Aspects Versus Position:**

8

**1-8 Gate Keeping:**

10

**1-9 Camera Ready Changes:**

1. We'll add details of human rating methodology as suggested by two reviewers.

We designed a speaker verification task using a two-alternative forced-choice paradigm. Participants listened to audio pairs containing an original sample and its attacked counterpart. For each pair, they answered the question: "Are these samples uttered by the same speaker/person?" by selecting either "The same speaker/person" or "Different speakers/people." To ensure our results were robust, three independent raters assessed each pair. Due to ethical considerations, we didn't collect any demographic information from the raters.

2. We'll add more discussion on potential short-term solutions and long-term solutions as suggested by all three reviewers.

Legislation: This involves creating a legal framework to deter misuse. Key elements include establishing laws that mandate clear labeling of synthetic audio, criminalizing the creation and distribution of malicious deepfakes, and defining liability for platforms and creators.

Watermarking with Rolling Key: Passive detection methods will ultimately become unreliable. This necessitates a strategic shift towards proactive watermarking. However, given that static watermarking schemes is also vulnerable [1,2], we need a strategy analogous to key rotation in cryptography, where the watermarking method is periodically updated to prevent adversaries from developing exploits against a long-standing, static implementation.

Community Education: The most resilient defense is an informed public. This requires widespread media literacy campaigns to teach citizens how to critically assess online content, recognize the signs of manipulation, and understand the capabilities of modern AI.

[1] Audiomarkbench: Benchmarking robustness of audio watermarking.
[2] Watermarks in the sand: Impossibility of strong watermarking for generative models.

3. We'll incorporate all of our answers to other reviewer feedback.

**3-1 Review Response1:**

TLox

**3-2 Reaction To Review1:**

The review is insightful and to the point. The review is supportive of our position. The review mainly focuses on the position over the technical aspect. The review is inclusive and not gate-keeping.

**3-3 Review Response2:**

scPU

**3-4 Reaction To Review2:**

The review is insightful and to the point. The review is supportive of our position. The review mainly focuses on the position over the technical aspect. The review is inclusive and not gate-keeping.

**3-5 Review Response3:**

iSXe

**3-6 Reaction To Review3:**

The review is insightful and to the point. The review is supportive of our position. The review mainly focuses on the position over the technical aspect. The review is inclusive and not gate-keeping.

---

### Meta-Review · Area_Chair_6rR4 · 2025-09-16

**Rating:** 7
**Confidence:** 3

**Strengths:**

clear agreement that the position is sound. A multi-faceted approach is indeed required to manage deepfake audio

**Weaknesses:**

should have analyzed in the frequency domain

**Questions:**

As noted by reviewers, lack of detail about watermarking. What about game theoretic perspective

**Thoroughness:**

2

---

### Decision · Program_Chairs · 2025-09-26

Reject